# Feeding climate and biodiversity goals with novel plant-based meat and milk alternatives

Marta Kozicka [1] ✉, Petr Havlík [1], Hugo Valin [1], Eva Wollenberg [2,3], Andre Deppermann [1], David Leclère [1], Pekka Lauri[1], Rebekah Moses[4], Esther Boere[1,5], Stefan Frank [1], Chris Davis[4], Esther Park[4] & Noel Gurwick[6]

Plant-based animal product alternatives are increasingly promoted to achieve more sustainable diets. Here, we use a global economic land use model to assess the food system-wide impacts of a global dietary shift towards these alternatives. We find a substantial reduction in the global environmental impacts by 2050 if globally 50% of the main animal products (pork, chicken, beef and milk) are substituted—net reduction of forest and natural land is almost fully halted and agriculture and land use GHG emissions decline by 31% in 2050 compared to 2020. If spared agricultural land within forest ecosystems is restored to forest, climate benefits could double, reaching 92% of the previously estimated land sector mitigation potential. Furthermore, the restored area could contribute to 13-25% of the estimated global land restoration needs under target 2 from the Kunming Montreal Global Biodiversity Framework by 2030, and future declines in ecosystem integrity by 2050 would be more than halved. The distribution of these impacts varies across regions—the main impacts on agricultural input use are in China and on environmental outcomes in Sub-Saharan Africa and South America. While beef replacement provides the largest impacts, substituting multiple products is synergistic.

Despite accounting for less than 20% of the global food energy supply, animal source foods (ASFs) are responsible for the majority of negative impacts on land-use[1], water use[2], biodiversity[1,3], and greenhouse gas emissions[4] in global food systems[5,6]. In low-income settings higher ASF consumption is recognized as a means to better nutrition, especially for children[7]. However, overconsuming some ASFs (particularly red meat and processed meats) has been shown to have adverse effects on health[8,9] and is threatening public health primarily in high income countries, but also increasingly in countries across the development spectrum, notably China, Mexico and Brazil[10–13]. Given these challenges, it is becoming clear that encouraging the adoption of low-ASF diets will be an important component in meeting climate change mitigation targets[14–16], achieving health and food security objectives worldwide[17,18], and keeping natural resource use within planetary boundaries[19–22]. Nevertheless, ASF consumption globally is projected

to continue growing[23,24]. There has been some success in encouraging plant-based diets in high-income countries, where ASF consumption is highest—increasingly more consumers identify as flexitarian (eating meat occasionally)[25,26]; however, only a few percent are vegan[27,28]. This suggests that a global shift in diets away from ASF will be challenging[29,30] and may require a range of technological and policy interventions[31,32].

One such emerging technology that has the potential to reduce ASF consumption following a behaviorally viable path[31] is the development of novel plant-based alternatives for meat and milk (hereafter referred to as novel alternatives). These are foods made from plants, mycelium, or other non-animal based ingredients, but developed to mimic the taste and consistency of animal products[33]. Despite their novelty, as of 2020 they have already gained popularity, with plant-based alternatives accounting for 15% of the milk market in the USA

[1]International Institute for Applied Systems Analysis, Laxenburg, Austria. [2]Gund Institute, University of Vermont, Burlington, VT, USA. [3]Alliance of Bioversity and CIAT, Cali, Colombia. [4]Impossible Foods, Redwood City, CA, USA. [5]Institute for Environmental Studies (IVM), VU University Amsterdam, Amsterdam, The Netherlands. [6]USAID Center for Development, Democracy, and Innovation, Washington, DC, USA. ✉e-mail: kozicka@iiasa.ac.at

and 1.4% and 1.3% of the meat markets in the USA[34] and Germany[35], respectively. Whilst novel alternatives yield local environmental benefits[36,37], the environmental and social impacts of large-scale adoption in the context of the complex global food system[38,39] are less well understood. To date, three dynamic macro-level studies of novel alternatives adoption have been executed, and they have been either limited in scale and scope[40,41], or have considered only a limited set of impacts and potential future market developments[42]. A comprehensive dynamic analysis of the global dietary change with the novel plant-based ASF alternatives has not been undertaken.

We address this research gap and provide a system-wide assessment of a large-scale (global) substitution of the main ASFs (pork, chicken, beef and milk) with novel alternatives. We used sets of hypothetical plant-based 'recipes' (details in "Methods" and Fig. 1), designed to be nutritionally equivalent to the original animal-derived products (ensuring macro- and micro-nutrient profile- and protein quality-equivalency). Further, we selected realistic ingredients that could feasibly be produced within existing food manufacturing capabilities and for global production (to balance global and regional abundance). We developed an array of forward-looking scenarios of dietary changes until 2050, which allows for considering different aspects of possible novel alternatives' market development. We analyzed these scenarios using the Global Biosphere Management Model (GLOBIOM), an economic partial equilibrium model that integrates global agriculture, bioenergy, and forestry sectors and allows for the exploration of the potential for the carbon sink and biodiversity restoration under additional land-use policy measures[43,44]. We started with a reference (REF) scenario to project dietary developments globally, using country-level characteristics of food demand based on consumer preferences. We then explored the global impacts of the dietary change on indicators such as GHG emissions, land-use, biodiversity, food prices, and food security (Fig. 1). In all scenarios, projections of socio-economic changes and population growth are based on the Shared Socioeconomic Pathway 2[45] 'middle-of-the-road' (SSP2) (details in "Methods"). Throughout the analysis, we assumed the current climate.

## Results
### Global food systems impact
In the REF scenario, food demand globally is projected to grow between 2020 and 2050 predominantly driven by higher incomes and larger population assumed in SSP2. Average per capita consumption of all main ASF products increases, with demand for chicken (+38%) and milk (+24%) growing the most (Fig. 2a). Total demand for crops increases, especially because of higher feed use (Fig. 2b). Thanks to the assumed technological progress, productivity increases result in lower prices for crops, by 3.8% (Fig. 2c). These factors, combined with the higher incomes and reduced inequality assumed in SSP2, lead to improved food security in all regions—the global prevalence of undernourishment declines from 8.4% in 2020 to 3.8% in 2050 (−269M people) (Fig. 2d). These undernourishment levels for 2050 are higher than previously projected[46], however they are more in line with the observed increase in the undernourishment levels in the last years, which signified a reversal of over a decade-long progress against hunger[47]. If impacts of climate change are taken into account, these numbers could be significantly higher[48,49].

In the alternative scenarios, we gradually substitute projected ASF consumption with novel alternatives, starting after 2020, and increase the mix linearly to achieve the targeted level by 2050. Some novel alternatives' recipes are significantly more economically competitive than others within the same category (e.g. within "chicken"), and therefore become dominant, especially at lower levels of substitution (Fig. 2a). At the higher substitution rates, market forces distribute the pressure over more commodities, resulting in a slightly more diverse selection of recipes (Table S1 in Supplementary notes 1). Total demand for most crops decreases compared to REF due to

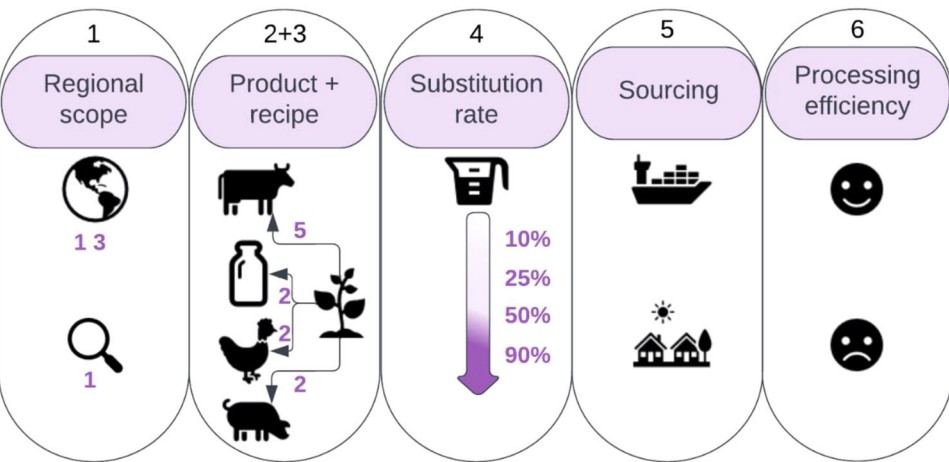

**Fig. 1 | Scenario dimensions of plant-based market development.** The substitution in the scenarios is defined along 6 dimensions. (1) We implemented the substitution globally or alternatively only within one of 13 macro regions (single-region scenarios): Brazil, China, Former Soviet Union, India, Middle East and North Africa, Other Asia, Other South America, Southeast Asia, Sub-Saharan Africa, Oceania, Canada, Europe, United States. (2) In each scenario, either a single ASF product (beef, chicken, pork, or milk) is substituted individually or all ASF products simultaneously. (3) There are five alternative recipes for beef and two alternative recipes for each of the other ASF products. In the single product scenarios, the recipe is selected exogenously and scenarios for each recipe were analyzed. In the scenario where all ASF products are substituted simultaneously, all recipes are permissible, and one is selected endogenously by the model based on the least cost of crop ingredients in the given GLOBIOM model economic region. (4) Four different levels of substitution were analyzed, corresponding to 10, 25, 50, and 90% ASF incremental as of 2020 substitution by 2050. The substitution was implemented by exogenously reducing the food consumption of ASF products by a percentage of the consumption projected in the REF scenario. (5) Sourcing of the novel alternatives ingredients is either global or local. In the locally sourced scenarios, imports are capped at a level equivalent to the level of imports in the REF scenario. The import cap is effective on the outside border of the 13 macro regions; trade within those regions is not restricted. (6) Utilization of the by-products of the conversion of primary crops into processed ingredients follows inefficient or efficient processing scenarios. In the inefficient scenarios, none of the processing by-products is assumed to find use in the global food system and therefore the entire amount of the primary commodity required to produce a given amount of the processed ingredient is utilized. In the efficient scenarios, all the processing by-products find use in the global food system and therefore only the amount of the primary commodity corresponding to the weight of the processed ingredient is utilized.

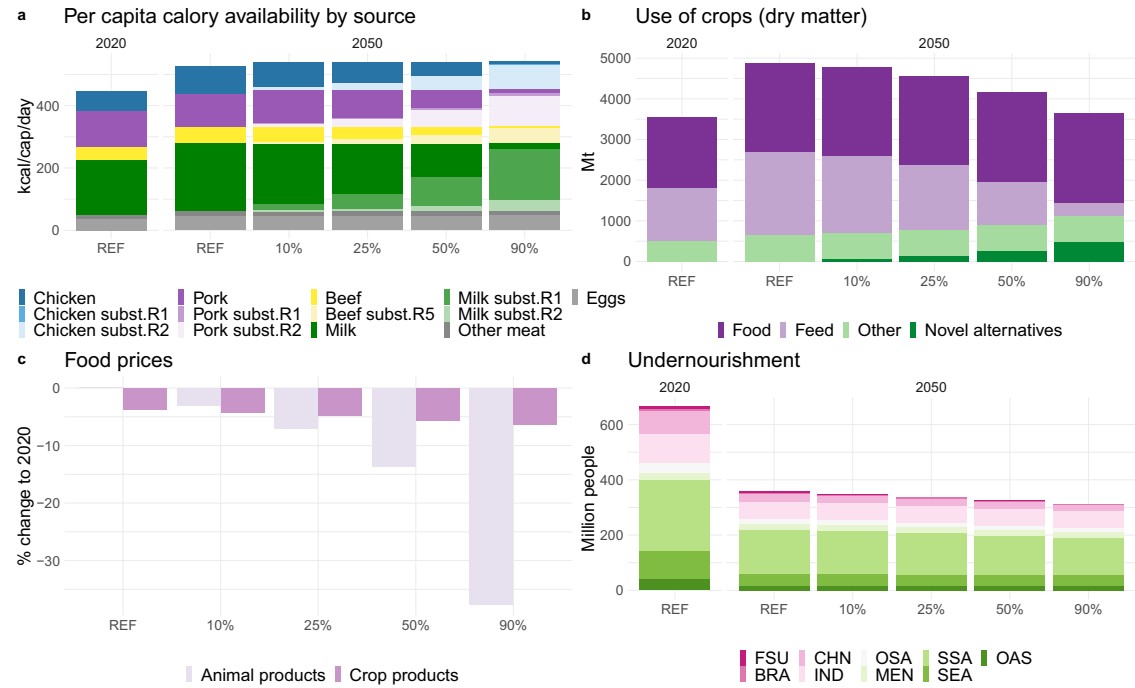

**Fig. 2 | Global food system indicators in 2020 and 2050 across scenarios (REF, 10%, 25%, 50% and 90% substitution globally).** Scenarios assume global substitution scope, all products, free trade sourcing, inefficient processing. Alternative scenario results are reported in the Supplementary Materials 4, 5 and 6. ASF products in figures (**a**) and (**c**) consist of the substituted products: beef, chicken, pork and milk and other ASFs: eggs and other meat (sheep and goat meat). On (**a**) novel alternatives recipes: five for beef (R1–R5) and two for each of the other products (R1–R2), included in the scenarios are detailed in "Methods". **b** Total use of crops in dry matter equivalent for food, feed, as components of the novel alternatives and other uses. In (**c**) the price changes were calculated using the Paasche index. The index excludes the novel alternatives prices, which are not considered in this study. In (**d**) regional abbreviations: Brazil (BRA), China (CHN), Former Soviet Union (FSU), India (IND), Middle East and North Africa (MEN), Other Asia (OAS), Other South America (OSA), Southeast Asia (SEA), Sub-Saharan Africa (SSA), Oceania (ANZ), Canada (CAN), Europe (EUR), United States (USA).

reduced demand for feed, which is not offset by the increased demand for production of the novel alternatives. The lower demand leads to further price reductions for both crops and ASFs and, through higher food availability, to improved food security. The changes are significant by 2050 even with 25% substitution levels. At 50% substitution, we see even more substantial shifts in most outcomes: total crop production is 20% higher in 2050 compared to 2020, while prices decline 14.1% for animal products, and 4.9% for crops. These price declines are within previously estimated ranges for dietary change impacts on food prices[50]. Undernourishment declines only moderately, to 3.6% (−31 million people compared to REF), with the largest impacts registered in Sub-Saharan Africa (−17 million people compared to REF).

Projected impacts depend on the processing efficiency associated with the replacement ingredients. The results presented above assume inefficient processing in which the totality of a plant is assigned to an input for a novel alternatives' recipe, e.g. a soy plant and a soy protein isolate (Supplementary notes 8). An efficient scenario is more economically likely and assumes a market for co-products (as occurs in agricultural processing today). If we assume high processing efficiency, crop use for novel alternatives drops by about 60% compared to the low-efficiency scenario, and consequently total demand for all crops decreases further compared to REF. However, the impact on other outcomes remains small, even negligible, at the lower substitution rates (Supplementary notes 4). This is due to rather small share of the crop use for novel alternatives as compared to the total crop use (Fig. 2b). At higher substitution rates, the impact can be more substantial, reducing crop use by even 8%. This means that our main results are based on conservative assumptions about processing and hence the benefits could be even higher if a more efficient processing scenario is realized.

## Global environmental impacts

In the REF scenario that assumes no plant-based dietary transformations, the agricultural sector exerts further pressure on natural resources from 2020 baselines until 2050 (Fig. 3a–d): agricultural area grows by 4% (+219 Mha) as it replaces forest and other natural land area (−5%, −255 Mha), nitrogen input to cropland grows by 39% (+59 Mt) and water use by 6% (+197 km³). Agriculture and land use related greenhouse gas (GHG) emissions increase by 15% (+1.1 Gt $CO_2$eq year⁻¹) and biodiversity, as measured by the Biodiversity Intactness Index, declines by 2.1%. These results are in line with the literature[51–53].

In the 50% substitution scenario, all impacts on natural resources decline significantly (Fig. 3a–d). Global agricultural area, instead of expanding, declines by 12%, while forest and other natural land area is 1% lower than in 2020. In total, 653 Mha of land is released from use as a result. Increase in nitrogen inputs to cropland as compared to 2020 is almost half of that projected in the REF scenario (+34 Mt relative to 2020). While the reduction in livestock population reduces the availability of cropland N inputs from manure by 6.4 Mt as compared to REF, savings in cropland N input needs are almost five times larger as the total cropland N input decreases by 27.8 Mt compared to REF. Water use declines by 10% (−291 km³) instead of increasing. Without accounting for any carbon sequestration on spared land, GHG emissions decline by 2.1 Gt $CO_2$eq year⁻¹ (31%) in 2050 (on average by 1.6 Gt $CO_2$eq year⁻¹ on 2020–2050). These values are in line with other estimates of a dietary change impacts found in the literature[41,54].

While agricultural input use declines almost proportionally to the substitution rate, land-use change and GHG emissions display nonlinear behavior. Substitution above 50% provides virtually no further reductions in deforestation (Fig. 3a) or land use $CO_2$ emissions (Fig. 3b). This 'saturation' occurs because at this substitution level, conversion of natural land to agricultural land ceases. The land use

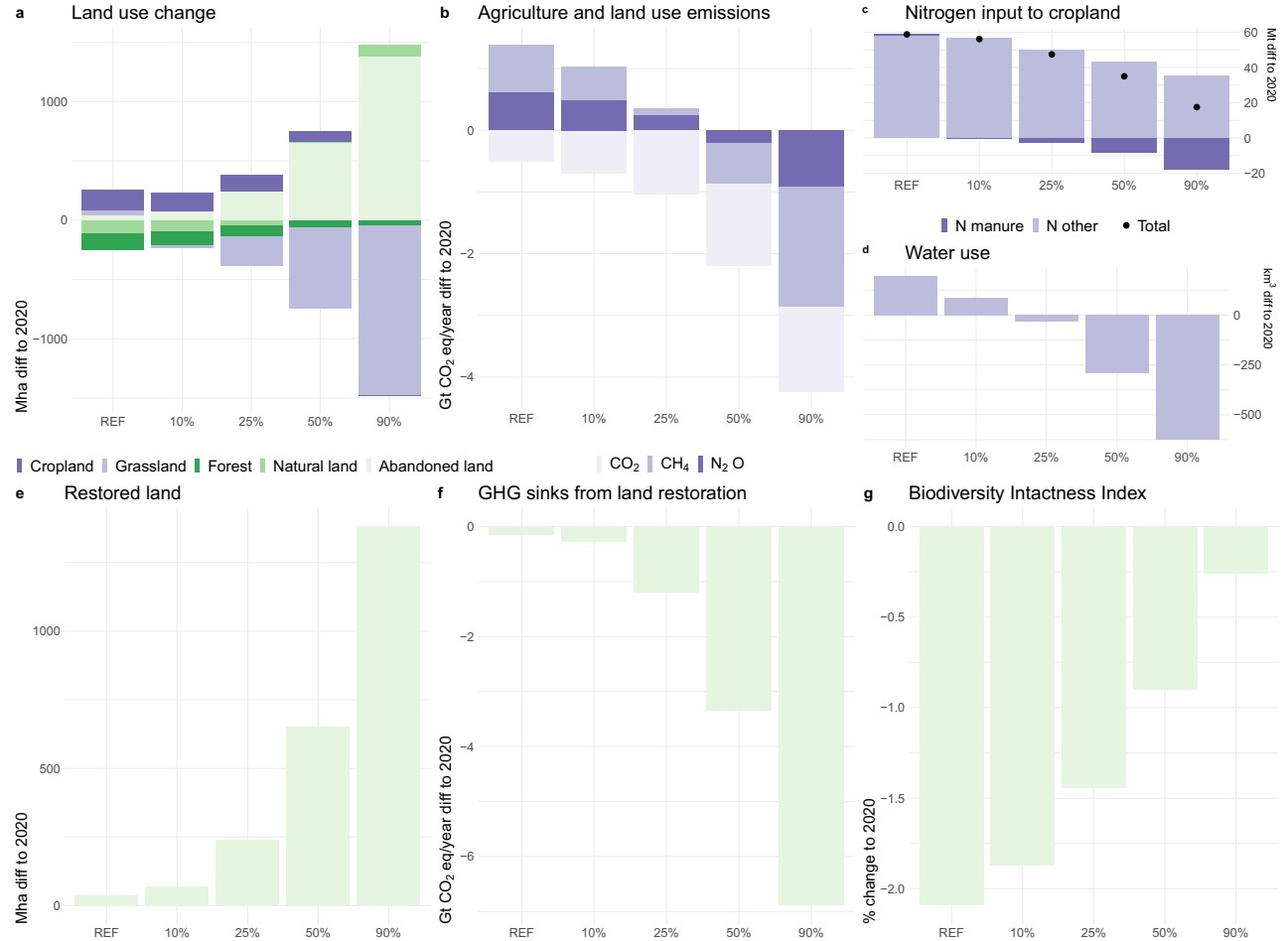

**Fig. 3 | Global environmental impacts in 2050 across scenarios (REF, 10%, 25%, 50% and 90% substitution globally).** Scenarios assume global substitution scope, all products, free trade sourcing, inefficient processing. Alternative scenario results are reported in the Supplementary Materials 4, 5 and 6. The top part of the panel (**a**–**d**) depicts results of the dietary change alone, while the bottom part (**e**, **f**, **g**) shows the impacts of implemented land restoration on the abandoned agricultural land. Land restoration was modeled as afforestation only within former forest ecosystems and with locally naturally occurring tree species.

change shifts to abandonment of agricultural land, which is higher at higher substitution rates. Similar effects have been observed in analysis of global substitution of beef[41]. Furthermore, we observe decreasing average livestock productivity and increasing emissions intensity, especially at 90% substitution, which is linked to the decreasing demand and thus prices of agricultural land, which encourage extensification of the ruminant systems, and due to the lower feed quality (energy density) also higher GHG intensity of the livestock products (Supplementary notes 2).

**Additional measure of land restoration—achieving the full environmental benefit**

If additional measures are taken to restore the agricultural land spared from livestock and feed production within forest biomes through afforestation with biodiversity-friendly management (Fig. 3e–g), the full environmental gain potential of the ASF substitution can be achieved. At 50%, carbon sequestration grows by 3.3 Gt $CO_2$ year$^{-1}$ in 2050 (1.5 Gt $CO_2$ year$^{-1}$ on average between 2020 and 2050) (Fig. 3f)— doubling the benefits already achieved through the reduction of land use emissions without such restoration. This gives 6.3 Gt $CO_2$eq year$^{-1}$ of all agriculture and land use emissions reduction compared to REF in 2050. This is close to the previously estimated land sector mitigation potential without bioenergy with carbon capture and storage (BECCS) of 6.8 Gt $CO_2$eq year$^{-1}$ in 2050[55]. Further increase in the substitution rate accelerates this potential—reaching 11.1 Gt $CO_2$eq year$^{-1}$ of total

agriculture and land use emissions reduction in 2050 in the 90% scenario (11.9 Gt $CO_2$eq reduction compared to REF).

Furthermore, the abandonment of agricultural land and its restoration in forest ecosystems allows to improve the state of biodiversity. As compared to 2020, the decrease in Biodiversity Intactness Index (BII) by 2050 is reduced to 0.9% in the 50% scenario (instead of 2.1% in the REF scenario), and to 0.3% in the 90% scenario (Fig. 3g), with a reversal of biodiversity loss achieved between 2030 and 2040 for the later scenario (Supplementary notes 3). We project an increase of 204 Mha (resp. 464 Mha) in land under restoration by 2030 as compared to 2020 in our 50% (resp. 90%) substitution scenario, which represents 58% (resp. 133%) of the 350 Mha targeted by the Bonn Challenge and 13-25% (resp. 29–57%) of the 810–1620 Mha needing restoration by 2030 to meet the ambition of the target 2 from the Kunming-Montreal Global Biodiversity Framework, if assuming that 20–40% of land area can be considered degraded[56]. It should be noted that the trends in BII accounts for time lags associated to restoration, and that restoration efforts in forest ecosystems continue beyond 2030 and almost triple by 2050, while restoration in non-forest ecosystem—not considered in this study—could further increase the biodiversity benefits from ASF substitution through restoration.

**Unequal distribution of the impacts regionally**

The model reveals vast differences in the impacts among regions. In the 50% scenario, China alone is responsible for 25% of the global crop

land abandonment (20.7 Mha) (Fig. 4a), and 22% of the direct agricultural (non-$CO_2$) emissions reduction (−0.5 Gt $CO_2$eq year$^{-1}$) (Fig. 4b). It also leads to reduced input use (Fig. 4c, d)−ca. 20% of the global water (−139 km$^3$) and nitrogen (−5.4 Mt). Sub-Saharan Africa exhibits the highest potential to reduce forest and natural land loss, by

−76.6 Mha compared to REF (37% of the global value), and reduce land use change emissions−by 0.4 Gt $CO_2$ year$^{-1}$ (47% of the global value). Furthermore, Sub-Saharan Africa, followed by Brazil, and Other South America, have the largest share of spared and restored land and corresponding carbon sequestration (Figs. 4a, b and 5). Even though these

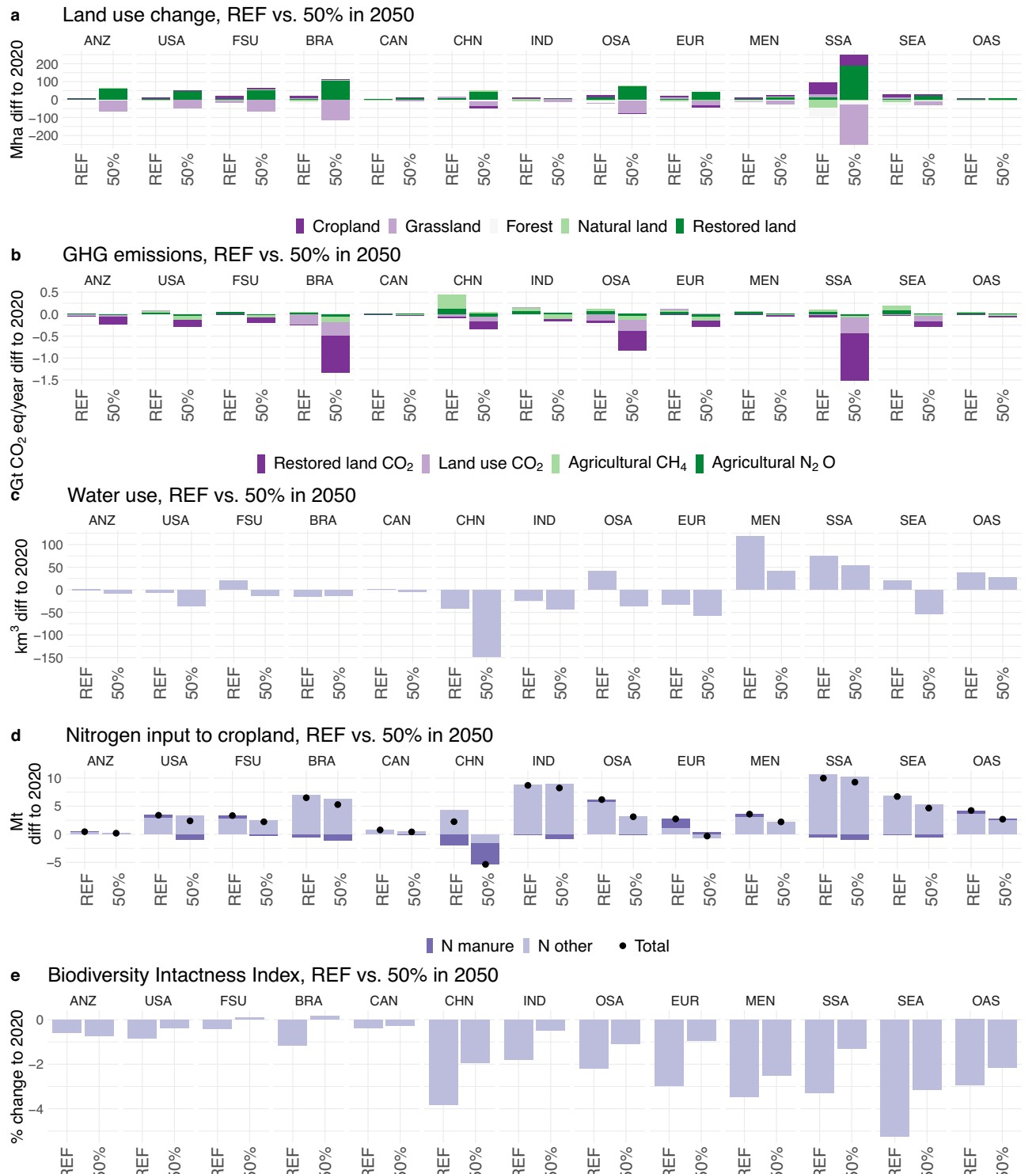

**Fig. 4 | Regional environmental impacts in 2050 across scenarios−REF and 50%.** Scenarios assume global substitution scope, all products, inefficient processing. Regional abbreviations: Brazil (BRA), China (CHN), Former Soviet Union (FSU), India (IND), Middle East and North Africa (MEN), Other Asia (OAS), Other South America (OSA), Southeast Asia (SEA), Sub-Saharan Africa (SSA), Oceania (ANZ), Canada (CAN), Europe (EUR), United States (USA). **a** presents land use change

assuming land restoration on the abandoned agricultural land. In figure (**b**), Land use $CO_2$ emissions refer to land use change related directly to agricultural use and do not take into account land restoration impacts, which are represented by Restored land $CO_2$ category. **c** presents change in water use and **d** change in nitrogen input to cropland. **e** presents Biodiversity Intactness Index impacts assuming land restoration on the abandoned agricultural land.

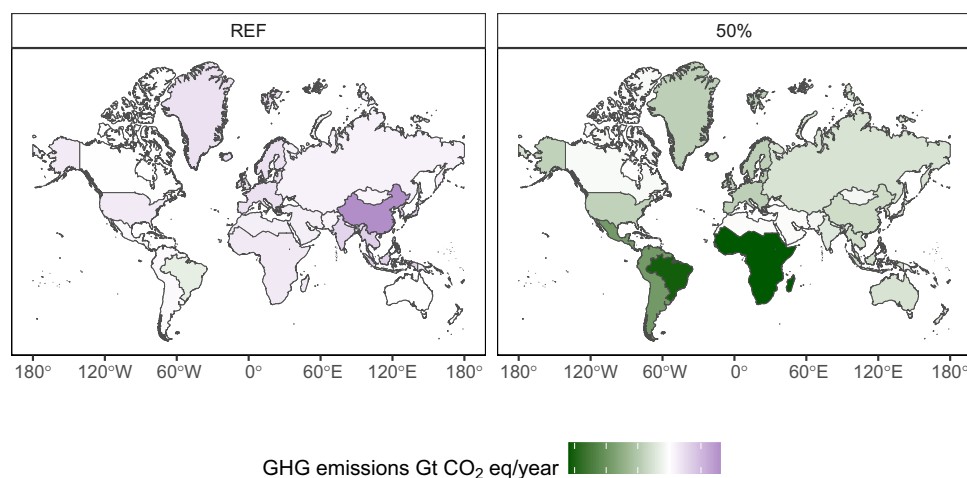

**Fig. 5 | Total agricultural and land use GHG emissions change by region in REF and 50% scenario.** Change in emissions between 2050 and 2020 is reported. Sinks from land restoration measures were included in the total emissions calculation. (REF) presents distribution of emissions in the reference scenario, while (50%) presents results under 50% global substitution scope, all products, inefficient processing.

three regions jointly consume only 22% of the world's total plant-based meat and 16% of milk, they reduce land use emissions by 2.7 Gt $CO_2$eq year$^{-1}$ (51% of the total global GHG emissions reduction) in 2050. Sub-Saharan Africa, together with China and South-East Asia are the biggest beneficiaries for avoided biodiversity intactness loss (BII decline lower by 2–2.1 percentage points compared to REF) (Fig. 4e).

Several factors drive such distribution of impacts among regions. First, regions in this study differ significantly with respect to population size and diets and resulting total demand for ASF and novel alternatives (Supplementary notes 1). As a result, China consumes globally the largest share of pork (45%) and chicken meat (15%); USA leads in beef consumption (15% of the world total); and India consumes the most milk (21% of the world total). These structural differences are important not only for scale, but also because the impacts significantly differ among products (Table S6, Supplementary notes 4). Beef substitution has the largest impact on GHG emissions, land use and biodiversity. About half of the reduction achieved by substituting all four animal products can be achieved by replacing only beef. However, chicken and pork substitution releases more cropland—clearly manifested in China. This result is in line with the literature[57,58]. In addition, there are synergies between products; substituting all products jointly has a larger impact than the sum of impacts from the individual product substitutions. Another critical driver of the regional differences is unequal productivity of the agricultural systems (Fig. S5, Supplementary notes 2). For example, Sub-Saharan Africa and Oceania have the most extensive beef production in 2050. Finally, regional impacts reflect adjustments in production and international trade of agricultural commodities (Supplementary notes 5). For example, big exporters of ASF, such as Brazil, see disproportionally larger impacts. Whether novel alternatives' ingredients are sourced locally or from global markets has a rather small impact on trade volumes in crops and almost no impact on ASF trade. Consequently, environmental impacts change only slightly between the two sourcing options, with the most pronounced impacts appearing for use of agricultural inputs.

Although global dietary changes deliver large benefits, it is likely shifts will not occur in all regions at the same pace. To better understand the impacts of having some specific regions moving faster than some others, we considered scenarios of single-region substitution. In this case, the impacts in that region are usually slightly smaller compared to the global scenario (Supplementary notes 6). Intuitively, the difference is larger for the regions with a high share of exports in the total production of the main ASFs, such as Oceania. This is because reduced domestic consumption has a rather small impact on

production in the region as the demand for exports continues. Furthermore, lower demand in one region leads to lower market prices and is partially offset by a rebound effect, with higher consumption in the rest of the world.

## Discussion

We show that substituting 50% of ASF with novel alternatives can lead to profound system-wide impacts. Unlike previous studies that assessed dietary changes with novel foods, in this study we considered a more realistic composition of the plant ingredients that would be used to produce novel alternatives and analyzed them in a dynamic system-wide global framework. Instead of growing by 15% in the REF scenario, agriculture and land use emissions decline by 31%. A large part of this decline comes from $CH_4$ reduction, which could have significant near-term climate mitigation benefits[59]. The result is comparable in relative terms to a previous analysis of replacing 60% of beef consumption in the USA with plant-based alternatives, which found agricultural emissions reduction in the USA by 13.5%[40]. The difference between the results could be explained by reduced ASF substitution coverage, relatively intensive beef production system in the region, and the rebound effect of higher demand globally. This means that proportionally, this study should report significantly smaller reduction in emissions as compared to ours.

Increasing the substitution rate beyond 50% led to only limited additional reductions of deforestation and associated emissions because substituting 50% of AFS with novel alternatives effectively eliminates land use change driven by those commodities. Yet, the abandonment of previously productive agricultural land offers the possibility of additional measures to restore land and sequester carbon, allowing for more than doubling of GHG emission reductions. As a result, the 50% scenario achieves 92% of the previously estimated global land sector abatement in 2050 consistent with 1.5 °C temperature increase strategy[60]. Restored land area in our 50% scenario reaches 58% of the Bonn Challenge targets and 13–25% of the target 2 from the Kunming-Montreal Global Biodiversity Framework for 2030[56]. Furthermore, we add to the literature by showing that this measure could more than half future declines in ecosystem integrity by 2050. This shows that together, the dual impact pathways of reduced deforestation at lower substitution levels and land restoration at higher levels could deliver results in line with ambitious climate change mitigation and biodiversity conservation goals.

Land-use policies and interventions would be necessary to achieve the full potential for carbon sequestration and biodiversity

conservation from restoration of abandoned agricultural land. Without these measures, new agricultural uses may prevail. In this study, we modeled afforestation with locally naturally occurring tree species, and only within former forest ecosystems. Consequently, our results present a lower bound estimate for carbon removals and biomass production from afforested land, but a mid-range estimate of the potential for biodiversity restoration. Further climate mitigation potential would come from sustainable use of the biomass from restored forests, and possibly from carefully selected sides for dedicated plantations, for bioenergy production. However, this aspect is outside of the scope of our paper. As a result, depending on the priorities and the policies in place, this balance could change. This would have especially strong implications in the regional analysis in temperate and boreal regions. While increasing tree cover in landscapes that lack trees or have low tree density could further increase carbon sequestration, those types of land use change interfere with a range of services provided by other ecosystem types. By restricting increased tree cover to previously forested landscapes in our model, we avoid these risks and conserve natural ecosystems[61] such as savannas[62,63]. Furthermore, we find that a large decrease in demand and consequently lower prices for livestock could lead to lower average livestock productivity and higher emissions intensity. As a result of this rebound effect, the net environmental benefit is smaller. This emphasizes the need for simultaneous interventions both on the supply and the demand side to accelerate the speed of transformation towards sustainable food systems.

The analysis of large-scale diet shifts is only one part of a larger challenge of the potential transition to a future with a more climate-friendly agricultural system, possibly with less livestock. Even though we find that the reduced demand for ASF results in higher food availability, which improved food security from 3.8% in REF to 3.6% in the 50% scenario, 326 million people globally remain undernourished in 2050. Broader economy impacts critical to human well-being, such as employment or farmers' livelihoods, which are not included in our analysis, would likely also be affected by the transition from ASF to novel alternatives[40]. Besides being a source of income for farmers, livestock has cultural roles and functions as a safety net and a diversification and risk-coping mechanism[64,65]. There are also complex interlinkages in mixed farming systems among feed, fertilizer, and soil quality[66] that need to be considered. Furthermore, we do not consider any potential negative impacts of price declines on the producers of agricultural commodities, which could be substantial[67]. Yet, environmental and climate change also represent great risks to livelihoods of those same producer groups, from local-scale impacts such as farm and forest community displacement, soil erosion, and water pollution to macro scale ecosystem service declines and climate change stressors[64,68]. Therefore, appropriate policy and management efforts should be developed to both prevent the environmental risks and to support farmers and other actors in the livestock value chain affected for a socially just transition. It is particularly important in the light of the recent setbacks to achieving food security globally[69], which might be further challenged by impacts of climate change[48,49]. One option is to support livelihood diversification or adaptation to the new novel alternatives value chains. Adoption of and operationalization of natural capital frameworks and ecosystem service payments that include environmental gains within economic reward structures can also prove useful for addressing livelihood transitions, biodiversity restoration, and climate change risks writ large.

A global introduction of novel alternatives has additional benefits compared to smaller-scale scenarios. Furthermore, substituting all ASFs simultaneously also shows synergies across the outcomes. However, regional substitution of individual products might be also highly effective in achieving particular objectives, especially if combined with regional strategies and purposeful selection of the 'recipes'. This is because of the heterogeneity of regions in this study with

respect to agricultural systems, diets, population sizes and natural resources, resulted in differences in the marginal impact of substitution, changes in trade, and the total volume of each of the products substituted. For example, substitution of beef in Brazil and in countries that serve as export markets for Brazilian beef and feed crops could initially reduce deforestation rates and, at higher substitution levels, open up land for reforestation[70]. Specific initiatives to incentivize reforestation on these abandoned grazing lands would increase the chances of achieving large-scale climate change mitigation benefits from dietary shifts in this country.

Although this study did not model the market for novel alternatives, there are many possible future scenarios of market development and novel alternatives adoption. We argue that the 50% substitution scenario is a realistic one, especially if the novel plant-based alternatives may be combined with traditional plant-based products and other novel meat substitutes, whether cell-based[71] or insect-based[72]. A major factor that will determine how these markets evolve is the price of the products. Currently, plant-based meat substitutes are generally more expensive than animal sourced meat, however the industry intends to reach price parity in some products even as soon as in 2024[73]. Novel alternatives are primarily processed food, often consumed away from home—a form of food consumption that developed rapidly in industrialized countries in the past century and is currently spreading to the rest of the world[74,75], accompanying urbanization and rising incomes[74,76], especially among youth. Furthermore, processed food and food away from home consumption is no longer confined to the urban middle class[76]. As world population growth is expected to come from the urban population of less developed regions[77], the large-scale shift in diets will either challenge human and environmental health or on the contrary create an opportunity to reformulate food consumption patterns towards sustainable diets, and introduce new products, such as alternative proteins[78]. Furthermore, the speed of substitution might follow different adoption curves or even tipping points, depending on the behavioral change patterns and the speed of technological progress[79,80]. Faster adoption would lead to even larger environmental benefits, especially in the form of larger emissions reduction. Policies could play a critical role on the one hand in fostering this transition (information campaigns, food labeling, public procurement, school programs, or even emission taxation etc.)[81], and on the other hand in ensuring equitable across regions and stakeholders distribution of the impacts that serve multiple sustainable development objectives. Decision-making hence needs to be based not only on the environmental and socio-economic benefits, but also on human health, and animal welfare impacts[82].

## Methods
We apply the Global Biosphere Management Model (GLOBIOM), a partial equilibrium economic model that integrates global agricultural, bioenergy, and forestry sectors[43,44] (Supplementary notes 9). The model is solved in a recursive dynamic manner in 10-year time steps from 2000 to 2050. Following McCarl and Spreen[83], market equilibrium is computed by allocating resources to production and processing activities while maximizing welfare (the sum of consumer and producer surpluses) subject to technological, resource, demand, and policy constraints[84]. Prices are calculated endogenously to balance supply, demand, and trade for each product and region. Agricultural policies are taken into account implicitly through adjustment of agricultural costs to reflect subsidies so that the marginal costs equal marginal benefits, as assumed by microeconomic theory. The policies are then assumed constant over the simulation horizon. Any change in the policies, such as in the subsidy levels, would cause a shift in profitability of the agricultural activities and would lead to a new equilibrium state, with new levels of supply, demand and prices.

Food demand projections are based on the interaction of three different drivers: population growth, income per capita growth,

response to prices. Population growth and income per capita growth are exogenously introduced in the model reference scenario. Price effect is endogenously computed, and the final demand in the model is therefore influence by some other assumptions on technology, natural resources, etc. that shape price patterns. Commodity markets and international trade are modeled at the level of 37 aggregate economic regions (Supplementary notes 7). The model's crop sector includes 18 major crops, and its production parameters are based on the biophysical crop model EPIC[85]. The livestock sector covers 7 major animal products, and it includes the representation of different International Livestock Research Institute/FAO production systems, agroecological zones, animal types, and management systems based on Herrero et al.[65]. The modeling of land use explicitly accounts for market dynamics related to scenario features, including both direct land use changes (e.g., domestic increases in the area of a crop in response a domestic demand increase) and indirect land use changes (e.g., within country re-allocations between crops and crop management systems, as well as changes in cropland allocation in other regions as mediated through trade).

GLOBIOM covers major greenhouse gas (GHG) emissions from Agriculture, Forestry and Other Land Use (AFOLU) based on IPCC accounting guidelines including $N_2O$ from application of fertilizer and manure to soils, $N_2O$ from manure dropped on pastures, $CH_4$ from rice cultivation, $N_2O$ and $CH_4$ from manure management, and $CH_4$ from enteric fermentation, and $CO_2$ emissions/removals from above- and below-ground biomass changes for other natural vegetation. In the nitrogen flows presented in this study (Figs. 3 and 4) we follow Chang et al. [53]. in estimating total nitrogen crop inputs (based on harvested crop nitrogen and nitrogen use efficiency) and manure nitrogen application (based on livestock excretion and manure management efficiency).

The forestry sector is represented by the biophysical Global Forest Model (G4M)[86,87]. G4M model provides the minimum and maximum constraint for the area of abandoned cropland and pastureland that can be afforested. Minimum afforestation is based on reference scenario projections at the regional level; maximum afforestation potential is based on suitability in terms of net primary productivity, at the pixel level. The estimate of the amount of carbon captured is based on the G4M model while the time dynamics of carbon density growth curves, differentiated by climate regions, are based on Humpenöder et al.[88]. The afforestation is modeled to take place with locally naturally occurring tree species. This allows us to assume improvements in biodiversity consistent with restoration of forest ecosystems. It might be possible to further increase biomass growth and carbon sequestration in certain areas with fast-growing plantations of energy grasses or fast-growing non-native tree species; however, this would have negative implications for biodiversity, besides being dependent on additional carbon capture and storage (CCS) technology. Biodiversity benefits could be potentially even higher if restoration would happen into non-forest ecosystems.

The afforestation is introduced in forest ecosystems, as constrained by gridded forest mask from the LUH2[89] dataset and afforestation potentials from the G4M model[86]. This allows us to avoid negative impacts of tree restoration on other natural ecosystems explored in some previous studies. Bastin et al.[61]. estimated that global tree restoration potential is about 900 Mha if current agricultural and urban areas are excluded. This study received a lot of attention and created a debate on the trade-offs of tree restoration benefits since majority of the restoration potential was located in natural land such as savanna where it would displace other valuable ecosystems[62,63].

Prevalence of undernourishment is calculated using three key factors: the mean dietary energy availability (kcal per person per day), the mean minimum dietary energy requirement (MDER) and the coefficient of variation of the domestic distribution of dietary energy availability in a country[90]. The food distribution in a country is assumed to obey a log-normal distribution, which is determined by the mean food calorie availability and the equity of the food distribution. The proportion of the population under the cut-off point (MDER) is then defined as the prevalence of under-nourishment. The calorie-based food consumption (kcal per person per day) output from the model is used for the mean food calorie availability. The future mean MDER is calculated for each year and country using the mean MDER in the base year at the country, adjusted for the MDER in different age and sex groups and future population demographics to reflect differences in the MDER across age and sex. The future equity of food distribution is estimated by applying the historical trend of income growth and the improved coefficient of variation of the food distribution to the future, such that the equity is improved along with income growth in future at historical rates up to the present best value (0.2). No risk of hunger for high-income countries where hunger is not currently reported is assumed.

Estimates of the intactness of local ecological communities are based on the Biodiversity Intactness Index (BII)[91], which measures the local compositional intactness of local communities as impacted by land use, relative to if the region were still covered with primary vegetation and facing minimal human pressures. To estimate the BII corresponding to the land use projected by GLOBIOM, we used the BII model derived from the PREDICTS database[92] as implemented in Leclère et al. [22], and complemented by an explicit accounting of the temporal dynamics of BII recovery under restoration actions based on Poorter et al. [93].

In the reference scenario (REF), trajectories of socioeconomic variables, income, and population, are based on the SSP2 'middle-of-the-road' development in the mitigation and adaptation challenges space[94]. It constitutes REF scenario because it assumes that the world follows a path in which social, economic, and technological trends do not shift markedly from historical patterns, with continuation of uneven development and income growth with persistent or slightly improving inequalities[45]. Throughout the analysis, we assume the current (historical) climate and don't apply any exogenous climate-specific shocks on crop yield, grassland yield, or animal productivity.

We compare the REF scenario results with the results of a suite of alternative ASF substitution scenarios developed specifically for this work. We implemented the substitution by exogenously reducing the food consumption of ASF products by a percentage of the consumption projected in the REF scenario. The reduction starts in the year 2030 and it is applied gradually to be completed (achieve the maximum level) by the year 2050. Simultaneously to the reduction in ASF consumption, we model an equivalent increase in the production and consumption of the corresponding novel alternatives. As a result, the total consumption of each ASF product and its novel alternative is always equal to the REF ASF consumption of that product. This assumes that the novel alternatives will substitute the ASF products at a 'one to one' substitution rate. The production of the novel alternatives is modeled as a process that requires crop feed inputs based on pre-defined recipes.

Each scenario is characterized by six elements: the substitution rate, the region where the substitution is applied, the ASF product being substituted, the recipe being used to produce the novel alternatives, the assumption on the sourcing of recipe crop ingredients, and the assumption on crop processing and the accounting of by-products.

Four different levels of substitution were analyzed, corresponding to 10, 25, 50, and 90% ASF substitution by 2050.

We implemented the substitution globally or alternatively only within one of 13 macro regions (single-region scenarios): Brazil, China, Former Soviet Union, India, Middle East and North Africa, Other Asia, Other South America, Southeast Asia, Sub-Saharan Africa, Oceania, Canada, Europe, United States. A mapping of GLOBIOM model regions into the 13 macro regions can be found in the supplementary material.

**Table 1 | Composition of recipes**

| Product | Recipe name | Ingredient 1 | Share | Ingredient 2 | Share | Ingredient 3 | Share | Oil ingredient share |
|---------|-------------|--------------|-------|--------------|-------|--------------|-------|----------------------|
| Beef | B1 | Soy protein isolate[a] | 20% | Sweet potato, dried[b] | 6% | | | 20% |
| | B2 | Rapeseed protein isolate[a] | 20% | Sugarcane (cane syrup)[b] | 6% | | | 20% |
| | B3 | Potato protein isolate[a] | 15% | Peanut flour[a,b] | 15% | | | 20% |
| | B4 | Soy protein isolate[a] | 10% | Potato protein isolate[a,b] | 6% | Wheat protein concentrate[a,b] | 4% | 20% |
| | B5 | Soy protein isolate[a] | 20% | Cassava, raw[b] | 10% | | | 20% |
| Chicken | C1 | Soy protein isolate[a] | 13% | Chickpea protein[a] | 13% | Wheat flour[b] | 10% | 8% |
| | C2 | Soy protein isolate[a] | 20% | Sweet potato, dried[b] | 10% | | | 8% |
| Pork | P1 | Beans, dry[a] | 15% | Soy protein isolate[a] | 13% | Sorghum flour[b] | 4% | 20% |
| | P2 | Soy protein isolate[a] | 20% | Wheat flour[b] | 4% | | | 20% |
| Milk | M1 | Soy protein concentrate[a,b] | 7% | Sugarcane (cane syrup)[b] | 2% | | | 3% |
| | M2 | Rapeseed meal[a,b] | 7% | Wheat protein concentrate[a] | 3% | | | 2% |

The M2 recipe would need to be supplemented.
[a]Protein role.
[b]Carbohydrate/binder role.

In a single-region scenario, the dietary change is implemented in one region, while the rest of the world has the same diet as in REF. This setting allows for studying, among others, impacts of changes in demand in one region on consumption in other regions and the net impact of these changes on the studied outcomes. It is expected that lower demand in one region will lead to lower prices, which are transmitted through markets to other regions. Due to lower prices, consumption in the rest of the world might increase, resulting in the net smaller decrease in global consumption and smaller benefits to the environment and other objectives. This outcome is referred to as a rebound effect.

In each scenario, either a single ASF product (beef, chicken, pork, or milk) is substituted individually or all ASF products are substituted simultaneously. There are five alternative recipes for beef and two alternative recipes for each of the other ASF products (Table 1). In the single product scenarios, the recipe is selected exogenously and scenarios for each recipe were analyzed. In the scenario where all ASF products are substituted simultaneously, all recipes are permissible, and one is selected endogenously by the model based on the least cost of crop ingredients in the given GLOBIOM model economic region.

Recipe development sought to balance global and regional abundance, efficiency, nutritional quality, and functional role of crop-derived ingredients in dietary replacement of livestock analogs.

Three primary categories were identified as delivering requisite functional attributes of recapitulating meat using crop-derived ingredients: proteins, binders (carbohydrates), and oils. All GLOBIOM crops were assessed against these categories as potential ingredient sources, ranked as high-, medium-, and low-potential and only the medium- and high-potential crops were retained.

Ensuring that the macro- and micro-nutrient profile, as well as protein quality, was consistent with that of the animal analog, we began with a high or medium protein source, and we calculated the amount (in %) of the crop required to reach approximate parity with the animal product being replaced. Next, a high/medium fat source was selected and the % use was calculated, again to meet parity with the animal analogue. Assuming that water would make up approx. 50–60% of meat recipes and approx. 90% of milk recipes, a high/medium carbohydrate crop was selected to make up the remainder of the recipe. If the carbohydrate source also contributed significant protein or fat, the % uses of the other protein and/or fat ingredients were adjusted to ensure that the total macronutrient contents of the recipes matched those of their animal analogs. Next, to ensure that protein quality was not sacrificed, total amino acid content for each recipe was calculated to determine its Amino Acid Score (AAS). The goal was for the AAS to match or exceed that of the animal analogue by achieving as close to a score of 1.0 (truncated) as possible. For

reference, the AAS for beef is 0.94[95]. These protein quality scores were not corrected for digestibility in this exercise. However, all recipes achieved an AAS between 0.9 and 1.0, except for the M2 milk recipe, for which the ability to supplement with additional amino acids was assumed. Table 1 below shows the composition of the recipes.

Each of the recipes calls for the inclusion of a vegetable oil. Six different oil crops are represented in the GLOBIOM model (soybean, rapeseed, sunflower, palm, groundnut, and cottonseed) and the respective oils were considered to be functionally equivalent for the purposes of recipe composition. For simplification, we exogenously assigned one vegetable oil in each of the 37 GLOBIOM model regions, and the choice of the vegetable oil varies between scenarios of global vs. local sourcing of novel alternatives crop ingredients.

Our recipes allow for the use of more diverse ingredients and are largely agnostic to current recipes on the market, in favor or hypothetical and realistic recipes that could be sourced, manufactured, and scaled globally. One or more modeled recipes are similar to the Impossible Burger and Beyond Burgers which are available in LCA studies[96,97], because the ingredients met the criteria nutritionally, functionally, and are within the GLOBIOM crop inventory.

For the global sourcing scenarios, we identified the most abundant oil in each market (whether or not produced locally, imported, or produced from imported crops) using statistics for the latest available year[98,99]. For the local sourcing scenarios, in each market we assigned the most abundant oil produced from locally grown crops, again based on latest available statistics. Table S18 (Supplementary notes 8) in the supplementary material shows the vegetable oil choice in each region and scenario setting.

Furthermore, in the locally sourced scenarios, imports are capped at a level equivalent to the level of imports in the REF scenario. This is to simulate the reliance on locally produced inputs into the production of novel alternatives. The import cap is effective on the outside border of the 13 macro regions; trade within those regions in not restricted. In the global sourcing scenarios, no changes or restrictions are applied to the reference scenario trade conditions and quantities. We do not consider trade with the final products of the novel alternatives, which would include all the additional costs, such as labor, and could potentially change the trade flows.

The novel alternatives recipes were constructed using processed ingredients such as flours and protein concentrates. In contrast, the GLOBIOM model generally follows the convention used by FAOSTAT and operates with quantities of primary crop equivalents. It was therefore necessary to convert the recipe ingredients into the corresponding primary crop equivalents. The supplementary material shows a detailed description of the processing conversions and yields of processed products and by-products.

Different assumptions that can be taken about the fate and potential utilization of the by-products of the conversion of primary crops into processed recipe ingredients give rise to two alternative sets of (processing) scenarios (Supplementary notes 4): inefficient and efficient. In the so-called inefficient scenarios, we assume that none of the processing by-products will find a use in the global food system and we therefore account for the entire amount of the primary commodity which is required to produce a given amount of the processed ingredient. In the efficient scenarios, we assume that all the processing by-products will find a use in the global food system; we therefore account for only the amount of the primary commodity corresponding to the weight of the processed ingredient.

We present these alternative approaches as an envelope of possible outcomes; depending on the specific ingredient, by-product, and crop, either the efficient or the inefficient assumption might be closer to reality. Today, several ingredients found in novel alternative recipes are byproducts from grain and protein processing, indicating that some degree of efficiency is likely, whether monetization of coproducts follows or is preceded by the ingredient's use in novel alternatives.

Our analysis does not consider the production costs of novel alternatives other than the costs of the crop ingredients mentioned earlier. Production costs would shape prices of the novel alternatives and hence decide about the level their adoption. We impose exogenously the level of production and consumption of novel alternatives and the substitution of ASFs is not affected by the relative prices of animal products and alternatives. We can expect that over time with scale and technological advances, the production costs of the novel alternatives would decrease and that product quality would increase - in line with the assumption of the increasing substitution over time.

The recipes were constructed to match the nutritional profile of typical animal product commodities; however, the nutritional profile of animal products varies regionally. In our analysis, we assume that the novel alternatives have the same regionally differentiated calorie content as the animal products they replace.

Besides meat and milk, the livestock sector also supplies animal fats for human consumption totaling approx. 14 million tonnes globally[98]. In the analysis presented here, we do not account for this quantity of animal products which would presumably need to be substituted with vegetable oils should a large-scale reduction in animal herds take place.

Finally, GLOBIOM is a partial equilibrium (PE) model, which means that the relevant sectors (agriculture, forestry and bioenergy) are represented in detail, however other economic sectors are not included, or only represented through an external variable (e.g. price of fertilizer). GLOBIOM assumes that the economy outside land use sectors evolves independently from the policies assessed in the model, following a ceteris paribus approach.

### Reporting summary

Further information on research design is available in the Nature Portfolio Reporting Summary linked to this article.

## Data availability

The GLOBIOM output data generated in this study, the R code and the supporting files for visualizing the results, have been deposited in the Zenodo database under accession code https://doi.org/10.5281/zenodo.8169317[100].

## Code availability

The Global Biosphere Management Model (GLOBIOM) documentation, links to GLOBIOM resources, GAMS script descriptions and dependency links that match the Trunk version of the GLOBIOM model are provided in a GitHub repository at https://iiasa.github.io/GLOBIOM/introduction.html.

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

## Acknowledgements

This work was implemented in partnership with Impossible Foods and Limestone Analytics. This study was supported by USAID, by the European Union under grant agreement number 101060483 - SWITCH (https://switchdiet.eu/), and as part of the CGIAR Research Program on Climate Change, Agriculture and Food Security (CCAFS), which is carried out with support from the CGIAR Trust Fund and through bilateral funding agreements. For details, please visit https://ccafs.cgiar.org/donors. The views expressed in this document cannot be taken to reflect the official opinions of these organizations. Special thanks to Miroslav Batka for providing valuable support with data collection, analysis and modeling, Michael Wögerer, Leopold Ringwald and Ipsita Kumar for advice on the figures design and to Scott Spillias for editorial support.

## Author contributions

P.H., H.V., E.W. and R.M. designed the study. A.D., D.L., P.L. carried out the GLOBIOM modelling with help from H.V., S.F., M.K. and E.B. M.K. performed the analysis and wrote an initial draft. L.W., P.H. and D.L. provided substantial inputs to the initial revisions to the draft. All authors contributed substantially to the interpretation of the results and to the final text.

## Competing interests

R.M., C.D., and E.P. declare the following competing interests: they were or are employees of Impossible Foods Inc., which is a company that develops plant-based substitutes for meat products. The remaining authors declare no competing interests.
