## [Peer Review File · Nature Communications]

Reviewers' Comments:

Reviewer #1:

Remarks to the Author:

Overall, I think this is important and interesting work. There is a good scope and the models used appropriate for the analysis. I would like to see more. However, in my opinion more has to be done to make the modelling assumptions, scenarios and model interconnections clear. I've divided my response by section of the paper and added some comments on the figures and writing below.

Introduction

The introduction is clear but the final paragraph on the methods could benefit from some further description. For example, it would be great to know what constitutes the BAU. In fact, it is based on SSP2 as we learn half way through the methods section - I would say SSP2 is not really BAU but if you want to use it as BAU it's important to defend in some way.

For those uninitiated with GLOBIOM it would be good to explain what it covers in terms of rebounds (more comments on this below because they only appear in the very last paragraph of the results), land allocation (based on yields and land prices I understand) and what it does not cover (in sectors). You don't want to overwhelm here as it is the end of the introduction, but it would be tremendously helpful for the general reader and you can add much more detail in the SI (more on that below).

Similarly, I think it would be good to explain here that you assess different recipes for the replacements.

Results

It's unclear from the start which results are exogenous drivers of the model and from which source, whether they are from the SSP. A good example is global food demand which I think may be driven by the pathway.

I would highlight a note of caution throughout the paper as current trends in food insecurity are not good and climate impacts may put a further dent into food availability. You do discuss the fact that climate feedbacks are ignored in the methods but I think it merits far more discussion in the main paper discussion. Line 97 suggests that the prevalence of undernourishment declines. Your statement that these levels are 'in line with recently revised statistics on undernourishment' I think applies to the point that undernourishment is higher than previously thought, but the inattentive reader could assume you are referring to the estimated reductions in 2050. This is all important because the model incorporates the prices of food and in this scenarios there are fewer relative pricing difference between animals and plants than in the real-world situation of increasing climate impacts on food systems.

You mention in line 100 that there is a linear substitution – how reasonable is this and do you have any potential evidence to cite for this assumption? On the one hand you could argue that some of these substitutions are technology based or behavioral based and might follow different adoption curves (or even tipping points), or you could argue that food culture change is hard. Some interrogation of this important assumption would help the reader understand the implications and results of the modelling.

In line 109 you mention the increase in crop production. I just want to warn a bit of modelling humility here given the very uncertain climate impacts on food systems which are coming earlier than previously modelled. Perhaps you could make a bigger point of this at the start of the results and note that the discussion below is predicated on the SSP drivers and not potential feedbacks.

In line 115 you explain that all the results so far assume inefficient processing. Why would we assume that? Why is that the first set of results you produce? In a world where we are seeing rising prices albeit with uncertain contributions from short- and long-term factors (war, climate, water scarcity) wouldn't we assume efficient processing with lower loss? (even if the model suggests that some prices decline).

In the paragraph at 152 you describe the land use emissions and mention they saturate. It may be worth clarifying here that this is when you assume no use of freed lands for land sparing which you turn to next. Or you can state at the beginning of the paragraph 'without accounting for any carbon sequestration on spared land we find that...'

In line 164 you discuss land restoration and carbon sequestration but put this in terms of a per year calculation rather than a total potential. My understanding of these types of models is that the trajectory of sequestration rates is highly uncertain, and you are on much safer ground by looking at the cumulative amount at saturation of the restored system. Could you please defend this annualized number either here or in the method?

In the paragraph at line 217 you discuss a regional shift, but it is unclear what the assumptions are here, and they are not described in the methods. A quick line somewhere that you isolate one region and apply the main assumptions would be good. I would also question whether this is really isolating the regional impacts as with these sorts of partial equilibrium models there may be important price-based factors that have regional impacts in the global model over and above those seen if you just model one model in isolation. It would be good to clarify this distinction here or in the methods.

At the end of this paragraph, you mention rebound effects for the first time – right at the end of the results. I would have expected description or discussion of this before now in the global assessment.

Discussion

The comparison in line 235 onwards is confusing. This is a comparison between global shifts to ASF with 50% against US only at 60% of beef? Could this comparison be made clearer?

In Line 267 I think you have to be clear about what you find and what is driven by the exogenous variables of the scenario when it comes to undernourishment.

Method clarifications

I would really appreciate a full description of all the interconnections of the model and the assumptions and implications. I was expecting quite a bit more detail in the SI but couldn't see anything.

To give an example: I am not clear about what carbon pools are included with the land restoration approach – is soil carbon included? To what depth? Is the land restoration to the potential vegetation before clearing? Some more details are required.

Further, you mention that you do not consider the production costs, what difference does this assumption make? Especially for the protein concentrates – here it seems like technological learning would make a difference, but it is not discussed at all.

An overview of the scenarios used in modelling is so very important, we only find that the BAU is based on SSP2 at the end of a different paragraph on biodiversity. I think this needs bringing out and describing fully – overall can you give an overview of which exogenous variables you used and where they come from (perhaps as plots of these variables and discussion in the SI). You have population and demand at the start of the SI but I'd like to see more, including the nourishment results that fall out from this. This is crucial for understanding what is driving the results. A list of assumptions for the model connections in the SI would also be great.

In terms of writing there are a few spots for improvement, a selection of what I noticed:

Very long sentences in the opening paragraphs.

Line 39 reads quite confusingly "further grow"

Some words such as "in the current study" in line 55 are redundant

Could you take another scan?

Figures:

There are notes at the end of each figure, it is unclear if these are figure captions

The text of most figures are quite small

Perhaps you can consider the text used in figures a little more. For example in figure 3 you have plenty of space to write 'difference' out fully rather than the abbreviated 'diff'.

Figure 4 is exceptionally hard to read.

Further Notes:

I don't think nature journals allow direct claims of novelty but will leave that up to the editors.

Reviewer #2:

Remarks to the Author:

Overall Comments

This is a well written manuscript on an important topic. Getting a sense of the potential impacts of novel plant-based alternatives on the environment is critical to considering these technologies as potential pieces of an overall portfolio of new practices and policies to reduce the environmental footprint of food systems.

The methodology is sound, and is based on a well regarded global economic and environmental model.

I have a few questions around scenario construction that I think the authors could address by clarifying a bit on their assumptions.

1. Is the demand for plant-based alternatives an exogenous assumption that is introduced into GLOBIOM? Likewise, is the substitution/replacement of conventional ASFs exogenous? I believe that the increase in demand for plant-based alternatives and reduction in ASF are both exogenous assumptions in the scenarios run in GLOBIOM. If this is true it would be helpful to make that a little clearer in the description of the scenarios. It would also be useful to also recognize that these scenarios don't include potential competition between novel plant-based alternatives (i.e., the proportion of plant-based beef alternative at the end of the projection period is the same as conventional beef with respect to overall meat consumption in the baseline scenario).

2. The authors suggest that the recipes used in the study are "realistic". I am assuming this is based on their estimate of nutritional equivalence. It would be helpful to explain what nutrient(s) are being used to create this equivalence, as some vitamins (e.g., vitamin B12) are not found in plants. It would also be interesting if the authors could compare their recipes to some of the recipes of existing plant-based alternatives (e.g., ImpossibleBurger or BeyondBurger) which are available in LCA studies.

Hellar, M. C., & Keoleian, G. A. (2018). Beyond Meat's Beyond Burger Life Cycle Assessment: A detailed comparison between a plantbased and an animal-based protein source.

<http://css.umich.edu/sites/default/files/publication/CSS18-10.pdf>

Khan, S., Loyola, C., Dettling, J., Hester, J., & Moses, R. (2019). Comparative Environmental LCA of the Impossible Burger with Conventional Ground Beef Burger.

<https://impossiblefoods.com/sustainable-food/burger-life-cycle-assessment-2019>

3. Where are the plant-based alternatives being produced? Are they being produced in the country where they are consumed? Is there any international trade in plant-based alternatives? One could imagine that some countries/regions would have comparative advantages in the production of plant-based alternatives. This might not alter environmental numbers, but could have important economic implications that are very relevant to policy makers. I understand that this too may be outside the scope of this study, but it would be good to be clear about this. It might be another justification for the very interesting single region scenarios that the authors ran.

While the focus of this study is on the environmental impacts, the authors also highlight changes to the risk of undernourishment. I believe that these scenarios are keeping GDP constant. However, livestock production is an important source of livelihoods for many in LMICs. GLOBIOM is a partial equilibrium model, so it probably isn't possible to get an estimate on the changes in income that a reduction in ASF production would present to rural populations. Nevertheless, it

would be good for the authors to note this as a point of uncertainty and need for future research when presenting the results on undernourishment, as a loss of income to these potentially vulnerable populations would certainly have a bigger impact on undernourishment than the projected price changes.

Specific Comments

Line 141: Does the modeling take into account reduced sources of organic fertilizer due to reduced ASF production? If it doesn't the authors should state this clearly when presenting the projected changes in inorganic N demand.

Line 179: Is this due to reductions in feed production for poultry and pork?

Line 187 These three regions meaning SSA, Brazil, and OSA? If so, the fact they consume only 22% of plant-based meat, seems like the wrong comparison, given that Brazil and OSA are major exporters of beef, so a reduction in their production of beef would be expected to be greater than their share of consumption.

Reviewer #3:

Remarks to the Author:

General comments to the authors

The authors address a new, interesting and important topic that has not yet been extensively studied. They make a valuable contribution by considering and analyzing multiple dimensions of alternative scenarios and multiple impact areas, using well-established methods with novel applications. I was expecting to see a bit more explicit discussion about trade-offs given the complexity of the issue and the many dimensions considered.

More specific comments are noted below.

Specific comments

Page 2, Line 33: the health effects of overconsuming ASFs are complex and controversial, and should be stated with more nuance here, e.g. "overconsuming some ASFs (particularly red meat and processed meats) has been shown to have adverse effects on health in..."

P3, L71: nice consideration of different scenario dimensions. Are different socioeconomic and climate pathways also considered? (Noted later at P11, L351.)

P3, L95: need to say a bit about the assumptions regarding technological progress.

P9, L270: good to see these caveats acknowledged explicitly.

P10, L342: I was expecting to see more discussion of trade-offs throughout the paper – whether observed between results for different indicators (or regions), or potentially between modeled results and omitted factors as noted at P9, L270.

P11, L351: These assumptions about population, income and climate would be noted earlier in the main text around P3, L71. At L351, it would be helpful to reflect a bit on how important these omitted factors might be in affecting the results.

P11, L387: this paragraph starts by talking about ensuring consistency of macro- and micro-nutrients, but really talks only about energy and protein. That's fine if that's what the methodology allows, but the first sentence should be qualified accordingly.

Dear Reviewers,

Thank you for your valuable comments and suggestions. Below please see a detailed point-by-point response to them.

Please find attached a revised manuscript with all the changes highlighted, followed by a cleaned version without the change tracking. The line numbers used in the responses refer to the track changes version.

Best regards,
Marta Kozicka

REVIEWER COMMENTS

Reviewer #1 (Remarks to the Author):

Overall, I think this is important and interesting work. There is a good scope and the models used appropriate for the analysis. I would like to see more. However, in my opinion more has to be done to make the modelling assumptions, scenarios and model interconnections clear. I've divided my response by section of the paper and added some comments on the figures and writing below.

Introduction

The introduction is clear but the final paragraph on the methods could benefit from some further description. For example, it would be great to know what constitutes the BAU. In fact, it is based on SSP2 as we learn half way through the methods section - I would say SSP2 is not really BAU but if you want to use it as BAU it's important to defend in some way.

Thank you for this suggestion. We added some more details of the scenario composition in the introduction section (lines 86-88) and more comprehensive explanation in the methods section (lines 516-519), together with the justification of using SSP2. We believe that SSP2 is a good choice of the socio-economic trajectory because it assumes that the world follows a path in which social, economic, and technological trends do not shift markedly from historical patterns, with continuation of uneven development and income growth with persistent or slightly improving inequalities.

For those uninitiated with GLOBIOM it would be good to explain what it covers in terms of rebounds (more comments on this below because they only appear in the very last paragraph of the results), land allocation (based on yields and land prices I understand) and what it does not cover (in sectors). You don't want to overwhelm here as it is the end of the introduction, but it would be tremendously helpful for the general reader and you can add much more detail in the SI (more on that below).

Thank you for this suggestion. We included more details about land allocation, trade impacts and GLOBIOM limitations (what it doesn't cover) in the methods section (lines 453-467, 606-608, 632-633, 646-650). We also added detailed GLOBIOM description in SI.

Similarly, I think it would be good to explain here that you assess different recipes for the replacements.

We would like to draw your attention to the last paragraph of the introduction (lines 68-71), where we explain that we used sets of alternative hypothetical plant-based 'recipes'. Further, Figure 1 and its caption contain more detail about the use of recipes in the scenarios (lines 89, 95-98).

Results

It's unclear from the start which results are exogenous drivers of the model and from which source, whether they are from the SSP. A good example is global food demand which I think may be driven by the pathway.

Indeed, demographic and macroeconomic drivers come from SSP2, that is they are exogenous, and they are mostly responsible for the demand increase. However, food demand is not fully

exogenously driven – it also responds to prices which are endogenous in GLOBIOM. We improved the way the results are introduced so that it is clearer what are the results and what are they driven by, that is what are the exogenous assumptions in the scenario (lines 115, 135-136). Furthermore, the improved methods section (lines 441-446, 492-505) and additional SI introducing GLOBIOM in more detail should make it very clear about what is endogenous and what is exogenous in our modeling framework.

I would highlight a note of caution throughout the paper as current trends in food insecurity are not good and climate impacts may put a further dent into food availability. You do discuss the fact that climate feedbacks are ignored in the methods but I think it merits far more discussion in the main paper discussion.

Thank you for this suggestion. We highlighted this issue both in the results (lines 122-124) and in the discussion sections (lines 354-357, 380-385). “Therefore, appropriate policy and management efforts should be developed to both prevent the environmental risks and to support farmers and other actors in the livestock value chain affected for a socially just transition. It is particularly important in the light of the recent setbacks to achieving food security globally⁶⁷, which might be further challenged by impacts of climate change^{68,69}.”

Line 97 suggests that the prevalence of undernourishment declines. Your statement that these levels are ‘in line with recently revised statistics on undernourishment’ I think applies to the point that undernourishment is higher than previously thought, but the inattentive reader could assume you are referring to the estimated reductions in 2050. This is all important because the model incorporates the prices of food and in this scenarios there are fewer relative pricing difference between animals and plants than in the real-world situation of increasing climate impacts on food systems.

Indeed, this is correct. Thank you for this observation. We have corrected the statement to make it clearer that the PoU increase was in the last years, reversing the trend of PoU decline (lines 122-124), and likely resulting in higher PoU until mid-century. Our projections until 2050 are more in line with this currently stalled progress.

You mention in line 100 that there is a linear substitution – how reasonable is this and do you have any potential evidence to cite for this assumption? On the one hand you could argue that some of these substitutions are technology based or behavioral based and might follow different adoption curves (or even tipping points), or you could argue that food culture change is hard. Some interrogation of this important assumption would help the reader understand the implications and results of the modelling.

Thank you for this suggestion. We added to the discussion (lines 417-420):

“Furthermore, the speed of substitution might follow different adoption curves or even tipping points, depending on the behavioral change patterns and the speed of technological progress. Faster adoption would lead to even larger environmental benefits, especially in the form of larger emissions reduction.”

In line 109 you mention the increase in crop production. I just want to warn a bit of modelling humility here given the very uncertain climate impacts on food systems which are coming earlier than previously modelled. Perhaps you could make a bigger point of this at the start of the results and note that the discussion below is predicated on the SSP drivers and not potential feedbacks. Here the comparison is between BAU and a substitution scenario, which assesses only the impact of the dietary shift on food availability and it isolates any impacts of climate-related drivers. For more clarity about our climate impact assumptions, we added already in the introduction section information that we are using current climate assumption in all scenarios (line 88).

In line 115 you explain that all the results so far assume inefficient processing. Why would we assume that? Why is that the first set of results you produce? In a world where we are seeing rising

prices albeit with uncertain contributions from short- and long-term factors (war, climate, water scarcity) wouldn't we assume efficient processing with lower loss? (even if the model suggests that some prices decline).

We aimed at focusing on the most conservative assessment of the impacts of the dietary change, showing the lower bound of these impacts first. We acknowledge this in the last sentence of the paragraph and we discuss the major differences if the opposite is assumed (lines 153-155). All the results from the efficient processing assumption are reported in SI. The two versions could be treated as presenting a lower and upper bound of the outcomes.

In the paragraph at 152 you describe the land use emissions and mention they saturate. It may be worth clarifying here that this is when you assume no use of freed lands for land sparing which you turn to next. Or you can state at the beginning of the paragraph 'without accounting for any carbon sequestration on spared land we find that...'

Thank you for this suggestion. We added this clarification to the paragraph (lines 184-185).

In line 164 you discuss land restoration and carbon sequestration but put this in terms of a per year calculation rather than a total potential. My understanding of these types of models is that the trajectory of sequestration rates is highly uncertain, and you are on much safer ground by looking at the cumulative amount at saturation of the restored system. Could you please defend this annualized number either here or in the method?

We are reporting both, the value by the end of the projection horizon, in 2050, and the average value per year, which was calculated from the cumulative sequestration of the projection period and divided by the number of years. For example (lines 214-215): "At 50%, carbon sequestration grows by 3.3 Gt CO₂ year⁻¹ in 2050 (1.5 Gt CO₂ year⁻¹ on average between 2020 and 2050)". This annualized reporting allows us to compare our results to other AFOLU mitigation studies.

In the paragraph at line 217 you discuss a regional shift, but it is unclear what the assumptions are here, and they are not described in the methods. A quick line somewhere that you isolate one region and apply the main assumptions would be good. I would also question whether this is really isolating the regional impacts as with these sorts of partial equilibrium models there may be important price-based factors that have regional impacts in the global model over and above those seen if you just model one model in isolation. It would be good to clarify this distinction here or in the methods.

The single-region scenarios are not isolating the region as such, but they introduce the dietary change in the selected region, while allowing for trade between all the regions. We added clarification of this set of scenarios in the methods (lines 545-552): "This setting allows for studying, among others, impacts of changes in demand in one region on consumption in other regions and the net impact of these changes on the studied outcomes. It is expected that lower demand in one region will lead to lower prices, which are transmitted through markets to other regions. Due to lower prices, consumption in the rest of the world might increase, resulting in the net smaller decrease in global consumption and smaller benefits to the environment and other objectives. This outcome is referred to as a rebound effect. "

At the end of this paragraph, you mention rebound effects for the first time – right at the end of the results. I would have expected description or discussion of this before now in the global assessment. Thank you for this suggestion. Due to space limitations in the main body of the article, we included this discussion (as quoted above) in the methods section (line 552).

Discussion

The comparison in line 235 onwards is confusing. This is a comparison between global shifts to ASF with 50% against US only at 60% of beef? Could this comparison be made clearer?

This comparison is between the relative decline in emissions as a result of the substitution. We improved the explanation to make it clearer what we are comparing and why (lines 311, 313-314).

In Line 267 I think you have to be clear about what you find and what is driven by the exogenous variables of the scenario when it comes to undernourishment.

Thank you for this suggestion. We improved the statement putting more emphasis on the causality (lines 354-357). We also included details of the calculation method of the PoU indicator in the methods section (lines 489-505).

Method clarifications

I would really appreciate a full description of all the interconnections of the model and the assumptions and implications. I was expecting quite a bit more detail in the SI but couldn't see anything.

To give an example: I am not clear about what carbon pools are included with the land restoration approach – is soil carbon included? To what depth? Is the land restoration to the potential vegetation before clearing? Some more details are required.

Thank you for pointing this out. Indeed, these details were missing. We added (lines 459-467)

“GLOBIOM covers major greenhouse gas (GHG) emissions from Agriculture, Forestry and Other Land Use (AFOLU) based on IPCC accounting guidelines including N₂O from application of fertilizer and manure to soils, N₂O from manure dropped on pastures, CH₄ from rice cultivation, N₂O and CH₄ from manure management, and CH₄ from enteric fermentation, and CO₂ emissions/removals from above- and below-ground biomass changes for other natural vegetation.”

We provide additional information in the methods section (459-467) and more detailed description of GLOBIOM in SI.

Further, you mention that you do not consider the production costs, what difference does this assumption make? Especially for the protein concentrates – here it seems like technological learning would make a difference, but it is not discussed at all.

Thank you for this suggestion. We added to the discussion section (lines 417-420) “ Furthermore, the speed of substitution might follow different adoption curves or even tipping points, depending on the behavioral change patterns and the speed of technological progress^{79,80}. Faster adoption would lead to even larger environmental benefits, especially in the form of larger emissions reduction.”

We added more discussion in the methods section (lines 631-637): “Our analysis does not consider the production costs of novel alternatives other than the costs of the crop ingredients mentioned earlier. Production costs would shape prices of the novel alternatives and hence decide about the level their adoption. We impose exogenously the level of production and consumption of novel alternatives and the substitution of ASFs is not affected by the relative prices of animal products and alternatives. We can expect that over time with scale and technological advances, the production costs of the novel alternatives would decrease and that product quality would increase -- in line with the assumption of the increasing substitution over time.”

An overview of the scenarios used in modelling is so very important, we only find that the BAU is based on SSP2 at the end of a different paragraph on biodiversity. I think this needs bringing out and describing fully – overall can you give an overview of which exogenous variables you used and where they come from (perhaps as plots of these variables and discussion in the SI). You have population and demand at the start of the SI but I'd like to see more, including the nourishment results that fall out from this. This is crucial for understanding what is driving the results. A list of assumptions for the model connections in the SI would also be great.

Thank you for this suggestion. We now provide these details in the SI.

In terms of writing there are a few spots for improvement, a selection of what I noticed:

Very long sentences in the opening paragraphs. Corrected

Line 39 reads quite confusingly “further grow” Corrected to “continue growing”

Some words such as “in the current study” in line 55 are redundant Removed

Could you take another scan? Thank you for the suggestions to the style. We have carefully read the paper again to improve the writing.

Figures:

There are notes at the end of each figure, it is unclear if these are figure captions

The text of most figures are quite small

Perhaps you can consider the text used in figures a little more. For example in figure 3 you have plenty of space to write ‘difference’ out fully rather than the abbreviated ‘diff’.

Figure 4 is exceptionally hard to read.

Thank you for all these suggestions. We have made the improvements and we hope that the figures are now easy to read as a result. Unfortunately, we had to keep the abbreviation ‘diff’ – in some figures the full word would not fit into the frame with the larger font.

Further Notes:

I don’t think nature journals allow direct claims of novelty but will leave that up to the editors. We have removed the novelty statement.

Reviewer #2 (Remarks to the Author):

Overall Comments

This is a well written manuscript on an important topic. Getting a sense of the potential impacts of novel plant-based alternatives on the environment is critical to considering these technologies as potential pieces of an overall portfolio of new practices and policies to reduce the environmental footprint of food systems.

The methodology is sound, and is based on a well regarded global economic and environmental model.

I have a few questions around scenario construction that I think the authors could address by clarifying a bit on their assumptions.

1. Is the demand for plant-based alternatives an exogenous assumption that is introduced into GLOBIOM? Likewise, is the substitution/replacement of conventional ASFs exogenous? I believe that the increase in demand for plant-based alternatives and reduction in ASF are both exogenous assumptions in the scenarios run in GLOBIOM. If this is true it would be helpful to make that a little clearer in the description of the scenarios. It would also be useful to also recognize that these scenarios don’t include potential competition between novel plant-based alternatives (i.e., the proportion of plant-based beef alternative at the end of the projection period is the same as conventional beef with respect to overall meat consumption in the baseline scenario).

This is a correct understanding of the substitution assumptions. We have made it explicit in the Figure 1 caption where the dimensions of scenarios are introduced (lines 100-101). Furthermore, we have improved the corresponding paragraph in the methods section (lines 530-532) by adding more details and the discussion of the substitution assumption. Thank you for your suggestion.

2. The authors suggest that the recipes used in the study are “realistic”. I am assuming this is based on their estimate of nutritional equivalence. It would be helpful to explain what nutrient(s) are being used to create this equivalence, as some vitamins (e.g., vitamin B12) are not found in plants.

Indeed, we ensured that the macro- and micro-nutrient profile, as well as protein quality, was consistent with that of the animal analogue. We added this explanation to the introduction (last

paragraph – lines 68-70) and added a reference to the methods section, which provides a detailed explanation of the process of developing the recipes to insure this equivalency.

It would also be interesting if the authors could compare their recipes to some of the recipes of existing plant-based alternatives (e.g., ImpossibleBurger or BeyondBurger) which are available in LCA studies.

Thank you for this suggestion. We added this comparison in the methods section (lines 592-596). The recipes are very similar in their composition. For the Impossible Burger potato protein, coconut oil, sunflower oil, and soy protein were used - similar to our recipe B4. The Beyond Burger was made of pea protein, canola oil and coconut oil - which is different from our recipes since we did not consider pea protein as an ingredient.

“Our recipes allow for the use of more diverse ingredients and are largely agnostic to current recipes on the market, in favor of hypothetical and realistic recipes that could be sourced, manufactured, and scaled globally. One or more modeled recipes are similar to the Impossible Burger and Beyond Burgers which are available in LCA studies^{96,97}, because the ingredients met the criteria nutritionally, functionally, and are within the GLOBIOM crop inventory.”

Hellar, M. C., & Keoleian, G. A. (2018). Beyond Meat’s Beyond Burger Life Cycle Assessment: A detailed comparison between a plantbased and an animal-based protein source.

<http://css.umich.edu/sites/default/files/publication/CSS18-10.pdf>

Khan, S., Loyola, C., Dettling, J., Hester, J., & Moses, R. (2019). Comparative Environmental LCA of the Impossible Burger with Conventional Ground Beef Burger.

<https://impossiblefoods.com/sustainable-food/burger-life-cycle-assessment-2019>

3. Where are the plant-based alternatives being produced? Are they being produced in the country where they are consumed? Is there any international trade in plant-based alternatives? One could imagine that some countries/regions would have comparative advantages in the production of plant-based alternatives. This might not alter environmental numbers, but could have important economic implications that are very relevant to policy makers. I understand that this too may be outside the scope of this study, but it would be good to be clear about this. It might be another justification for the very interesting single region scenarios that the authors ran.

We study trade of the components of the novel alternatives using the trade dimension of the scenarios. The free trade setting allows for a free flow of all the commodities. (In GLOBIOM trade is modelled following the spatial equilibrium approach based on cost competitiveness and homogeneous good assumption that allows for tracing of bilateral trade flows between individual regions.) In the locally sourced scenarios, imports are capped at a level equivalent to the level of imports in the BAU scenario. This is to simulate the reliance on locally produced inputs into the production of novel alternatives. We found that whether novel alternatives’ ingredients are sourced locally or from global markets has a rather small impact on trade volumes in crops and almost no impact on ASF trade. It would be indeed interesting to consider trade with the final products, which would include all the additional costs, such as labor. This is however beyond of our modeling scope. It would be possible and relevant to study bilateral trade flows and how they change under local sourcing scenarios. We will leave this for the future research since this is beyond the scope of the current study. We added additional information to this matter to the methods section (lines 606-608).

While the focus of this study is on the environmental impacts, the authors also highlight changes to the risk of undernourishment. I believe that these scenarios are keeping GDP constant. However, livestock production is an important source of livelihoods for many in LMICs. GLOBIOM is a partial equilibrium model, so it probably isn’t possible to get an estimate on the changes in income that a

reduction in ASF production would present to rural populations. Nevertheless, it would be good for the authors to note this as a point of uncertainty and need for future research when presenting the results on undernourishment, as a loss of income to these potentially vulnerable populations would certainly have a bigger impact on undernourishment than the projected price changes.

This is an important point. We base our scenarios on SSP2, which assumes GDP growth. To improve the clarity, we have added this information in the introduction (lines 86-88, 115) and more details in the methods section (lines 442-446). We provide the discussion about the impacts of potential livestock reduction on the livelihoods and the ramifications for the food security in the discussion section (lines 380-385).

Specific Comments

Line 141: Does the modeling take into account reduced sources of organic fertilizer due to reduced ASF production? If it doesn't the authors should state this clearly when presenting the projected changes in inorganic N demand.

Thank you for this observation. We have taken the opportunity to slightly revise the reporting of the nitrogen use (lines 464-467). The current estimation follows Chang et al. (a paragraph with more details added to the methods section). We currently report total nitrogen crop inputs, as compared with the nitrogen fertilizer use in the previous version. As a result, the updated numbers are slightly larger. However, our updated estimates contain manure-sourced fertilization, which allows us to compare total crop nitrogen input decrease resulting from the lower crop production, with the manure-sourced N application. We show that the total nitrogen use reduction is much larger than the reduction in manure-sourced nitrogen.

Chang, J., Havlík, P., Leclère, D., de Vries, W., Valin, H., Deppermann, A., Hasegawa, T., & Obersteiner, M. (2021). Reconciling regional nitrogen boundaries with global food security. *Nature Food*, 2(9), 700–711. <https://doi.org/10.1038/s43016-021-00366-x>

Line 179: Is this due to reductions in feed production for poultry and pork?

Indeed, the impact is related to a large share of chicken and pork consumption in China. We have added explanation of this link in the text (lines 275-276).

Line 187 These three regions meaning SSA, Brazil, and OSA? If so, the fact they consume only 22% of plant-based meat, seems like the wrong comparison, given that Brazil and OSA are major exporters of beef, so a reduction in their production of beef would be expected to be greater than their share of consumption.

Indeed the distribution of impacts depends on several factors, such as trade and productivity of agricultural systems. We explain the reasons for these effects in the paragraph below Figure 4 (lines 282-283).

Reviewer #3 (Remarks to the Author):

General comments to the authors

The authors address a new, interesting and important topic that has not yet been extensively studied. They make a valuable contribution by considering and analyzing multiple dimensions of alternative scenarios and multiple impact areas, using well-established methods with novel

applications. I was expecting to see a bit more explicit discussion about trade-offs given the complexity of the issue and the many dimensions considered.

More specific comments are noted below.

Specific comments

Page 2, Line 33: the health effects of overconsuming ASFs are complex and controversial, and should be stated with more nuance here, e.g. “overconsuming some ASFs (particularly red meat and processed meats) has been shown to have adverse effects on health in...”

Thank you for pointing this out. We have changed the statement by adding this nuance (lines 38-40).

P3, L71: nice consideration of different scenario dimensions. Are different socioeconomic and climate pathways also considered? (Noted later at P11, L351.)

We consider only one socio-economic (SSP2) and climate (current climate) pathway in all scenarios in order to isolate the impact of the dietary change. We have added this information in the introduction so that it is clear for a reader from the beginning (lines 86-88).

P3, L95: need to say a bit about the assumptions regarding technological progress.

Technological progress assumed is consistent with the SSP2 pathway, which follows historical trends. We added this explanation to the methods section (line 516-519).

P9, L270: good to see these caveats acknowledged explicitly.

P10, L342: I was expecting to see more discussion of trade-offs throughout the paper – whether observed between results for different indicators (or regions), or potentially between modeled results and omitted factors as noted at P9, L270.

Thank you for this suggestion. We added more discussion of trade-offs and rebound effects. Furthermore, we added more discussion of different assumptions and possible impacts on the outcomes that are not covered by the GLOBIOM model. For example, we added discussion of observed decreasing average livestock productivity and increasing emissions intensity along the increasing substitution rates (Global environmental impacts subsection of the results and the discussion section lines 196-199, 345-350), expected impacts of climate change on food security (lines 383-385), inter-linkages in mixed farming systems among feed, fertilizer, and soil quality (lines 364-365), single-region substitution scenario rebound effects (lines 545-553) or the discussion of possible impact of the novel alternatives’ production costs that are not included in our modeling framework (lines 631-637).

P11, L351: These assumptions about population, income and climate would be noted earlier in the main text around P3, L71. Thank you for this suggestion. We have added this information in the introduction (lines 86-88).

At L351, it would be helpful to reflect a bit on how important these omitted factors might be in affecting the results. Thank you for this suggestion. In the methods section, we expanded the description of SSP2 and added discussion of the assumption of current climate (lines 514-521).

P11, L387: this paragraph starts by talking about ensuring consistency of macro- and micro-nutrients, but really talks only about energy and protein. That’s fine if that’s what the methodology allows, but the first sentence should be qualified accordingly.

In fact, there is a discussion of amino acids later in this paragraph (line 577-579), which are considered to be micro-nutrients. We would like to keep the formulation used.

Reviewers' Comments:

Reviewer #1:

Remarks to the Author:

The authors have done a good job of addressing my comments. I think it is sufficient for publication after addressing my comments below:

There seems to be some issues with line numbers. For example, additions in the methods section at 606 and 632 appear to have been made but these are in the references section.

It would also be great add the updated text in the response document to save having to scan through different lines in the manuscript (wherever possible, I'm aware that some additions can be very long).

To the specific points:

I will disagree that the SSP2 is BAU, and even more strongly this time. I would be careful with using any SSP in such a descriptive way as a BAU. SSPs do not reflect climate damage at all and so cannot really be considered BAU. You have defended it, but I would much rather say this is an SSP2 scenario, with all that entails, rather than calling it a BAU and giving the impression that this future is the business as usual we are on. This is important as many scientists and policy makers will not understand these distinctions. We have seen these problems with the use of RCP8.5 as a BAU. You can still say that these AP transitions result in reductions in undernourishment compared to an SSP2 middle-of-the-road scenario.

Further to this point, I would like further comment regarding food insecurity. I note that undernourishment drops to 2050 under the SSP2 scenario with no interventions. This is a similar result to those detailed here for food insecurity: <https://www.nature.com/articles/s43016-021-00322-9> which is at odds with current trends (it has been increasing for several years, and prior to COVID). Food models are showing increasing climate risks over time, and those models with extreme events are showing concerning results. I think it is important to provide further comment on this before isolating the results against the food transitions.

Thank you for the extra detail on GLOBIOM. I assume there is no inclusion of subsidies in the model? As you know agricultural prices are highly distorted by these which can have impacts on the model itself given it is price-based. I think this deserves some comment in the manuscript – perhaps somewhere in lines 308-325.

Otherwise, congratulations on the work!

Reviewer #2:

Remarks to the Author:

I think this is a well written article on an important topic. The authors responded to all the concerns I raised in the first round of review, and I believe that the manuscript is publishable.

One small edit in the abstract I would suggest including the definition of main animal products as was done in the introduction.

REVIEWERS' COMMENTS

Reviewer #1 (Remarks to the Author):

The authors have done a good job of addressing my comments. I think it is sufficient for publication after addressing my comments below:

There seems to be some issues with line numbers. For example, additions in the methods section at 606 and 632 appear to have been made but these are in the references section.

It would also be great add the updated text in the response document to save having to scan through different lines in the manuscript (wherever possible, I'm aware that some additions can be very long).

To the specific points:

I will disagree that the SSP2 is BAU, and even more strongly this time. I would be careful with using any SSP in such a descriptive way as a BAU. SSPs do not reflect climate damage at all and so cannot really be considered BAU. You have defended it, but I would much rather say this is an SSP2 scenario, with all that entails, rather than calling it a BAU and giving the impression that this future is the business as usual we are on. This is important as many scientists and policy makers will not understand these distinctions. We have seen these problems with the use of RCP8.5 as a BAU. You can still say that these AP transitions result in reductions in undernourishment compared to an SSP2 middle-of-the-road scenario.

We acknowledge that the use of the term 'business-as-usual' might be misleading and hence we have decided to change the name from BAU to the 'reference (REF) scenario', which does not imply continuation of the current trends, for example with respect to climate change. We made the corresponding changes throughout the manuscript and the supplementary materials.

Further to this point, I would like further comment regarding food insecurity. I note that undernourishment drops to 2050 under the SSP2 scenario with no interventions. This is a similar result to those detailed here for food insecurity: <https://www.nature.com/articles/s43016-021-00322-9> which is at odds with current trends (it has been increasing for several years, and prior to COVID). Food models are showing increasing climate risks over time, and those models with extreme events are showing concerning results. I think it is important to provide further comment on this before isolating the results against the food transitions.

In addition to already mentioned changes in the trends in food security, we have added an emphasis on the potential upward risk to PoU numbers due to climate change also in the results section (line 111-112): "If impacts of climate change are taken into account, these numbers could be significantly higher^{48,49}."

Thank you for the extra detail on GLOBIOM. I assume there is no inclusion of subsidies in the model? As you know agricultural prices are highly distorted by these which can have impacts on the model itself given it is price-based. I think this deserves some comment in the manuscript – perhaps somewhere in lines 308-325.

We have added explanation of the consideration of agricultural subsidies in GLOBIOM and the discussion about their potential impacts on the modelling results in the methods section. Line 379-384 "Agricultural policies are taken into account implicitly through adjustment of agricultural costs and subsidies so that the marginal costs equal marginal benefits, as assumed by microeconomic theory. They are assumed constant over the simulation horizon. Any change in the policies, such as

in the subsidy levels, would cause a shift in profitability of the agricultural activities and lead to a new equilibrium state, with new levels of supply, demand and prices. ”

Otherwise, congratulations on the work!

Reviewer #2 (Remarks to the Author):

I think this is a well written article on an important topic. The authors responded to all the concerns I raised in the first round of review, and I believe that the manuscript is publishable.

One small edit in the abstract I would suggest including the definition of main animal products as was done in the introduction.

Thank you for the suggestion – we added the list (pork, chicken, beef and milk) in line 21.